# *"People in my life just play different roles"*: A retrospective qualitative study of friendships among young adults who self-harmed during adolescence

**Delfina Bilello**[1,2,3]*, **Ellen Townsend**[4], **Matthew R. Broome**[1,2], **Stephanie Burnett Heyes**[1,2]

**1** School of Psychology, University of Birmingham, Edgbaston, Birmingham, United Kingdom, **2** Institute for Mental Health, University of Birmingham, Edgbaston, Birmingham, United Kingdom, **3** Department of Experimental Psychology, University of Oxford, Oxford, United Kingdom, **4** Self-Harm Research Group, School of Psychology, University Park, The University of Nottingham, Nottingham, United Kingdom

* delfina.bilello@psy.ox.ac.uk

## Abstract

Self-harm is a global public health concern presenting increasing rates in recent years, especially among young people. This population seldom access formal help, and typically rely on informal sources of support, mainly friends. The role, importance and meaning of friendships in the context of self-harm remains poorly understood, highlighting the need to explore young people's lived experiences. In the present study we conducted semi-structured retrospective qualitative interviews, prompted by the Card-sort Task for Self-Harm About Friends (CaTS-AF), to explore the experiences of 11 young adults (M=19.09; SD=0.70; M=2, F=9) who self-harmed during adolescence. Data were analysed using Reflexive Thematic Analysis (RTA). Three themes were developed which consider 1) the role of friendships in self-harm progression; 2) the role of self-harm in friendship evolution; and 3) the meaning of friendships in the context of adolescents' self-harm. The first two themes highlight the interdependent nature of friendships and self-harm, where these two experiences influence one another. Furthermore, not only are friendships shaped by self-harm, but they acquire specific meanings, dynamics and expectations within the context of the behaviour. Overall, friends are a key part of adolescents' self-harm, as sources of both risk and protection. It is essential to further integrate friendships, a developmentally significant aspect of adolescents' social experiences, into self-harm research and clinical practice.

## 1. Introduction

Self-harm, defined as any behaviour intended to harm oneself regardless of intent, is a global health concern with steadily increasing rates especially among adolescents [1]. According to the World Health Organisation, adolescence encompasses the period from 10–19 years old, while the term young people refers to individuals between 10–24 years old [2]. In the UK, around 39% of individuals aged 17–24 years old report having ever self-harmed, with prevalence rates greater among females [3]. Relative to rates of self-harm, the proportion of

**Data availability statement:** We have ethical restrictions in regards to data sharing given the sensitivity and potentially identifiable nature of the data collected during the project. Data consists of transcribed and potentially identifying qualitative interviews about young people's experiences of self-harm. Therefore, we have ethical and participants' restrictions which prevent public sharing of minimal data for this study. Data are available upon request from the University of Birmingham Ethics Committee via email (ethics-queries@contacts.bham.ac.uk).

**Funding:** Economic and Social Research Council (ESRC) PhD Studentship to Delfina Bilello.

**Competing interests:** The authors have declared that no competing interests exist.

young people accessing formal support for self-harm is substantially lower, such that support is rarely accessed or help-seeking may be substantially delayed [4,5]. Instead, the majority of young people who self-harm typically rely on their informal networks, primarily friends, who are known to become important sources of support [6,7]. Research suggests that friends can positively and negatively impact self-harm [8], yet more research is needed to gain a better understanding of the importance and role of friendships in the context of the behaviour.

## 1.1. Self-harm risk and protective factors: The role of friendships

Self-harm is a complex behaviour, thought to emerge over time from the interplay between genetic, biological, psychological, social and cultural factors [9]. Current theories such as the Integrated Motivational-Volitional Model of Suicidal behaviour (IMV; see [10,11]) emphasise that individuals transition through distinct stages of risk from vulnerability (pre-motivational stage), to self-harm ideation (motivational stage) to self-harm enaction (volitional stage), each influenced by distal and proximal factors. A key dimension observed across stages is the presence of social factors, including friendships.

Friendships are typically considered positive, voluntary and reciprocal informal relationships [12]. However, these evolve over time and follow different trajectories as well as acquiring distinct roles and characteristics across the lifespan, in line with individuals' developmental needs [13,14]. During adolescence, friendships gain importance as primary sources of companionship, emotional and social support, as well as being key influences on development and behaviour [15–17].

Friendships have primarily been conceptualised in research as either risk or protective factors for self-harm [8,18]. On one hand, evidence of the former recognises negative experiences with peers, including bullying, victimisation and interpersonal stress, as key factors increasing self-harm vulnerability [19,20]. A lack of peer support, perceived burdensomeness, and belongingness have been extensively explored as mediators and moderators of self-harm ideation [21]. Research has particularly concentrated on exposure to friends' self-harm as a key volitional factor for self-harm onset, which typically occurs in early adolescence [22,23].

Research on friends as sources of support is comparatively scarce [8,24,25]. Friendship quantity and quality have been observed to be key protective factors against self-harm both indirectly through increases in self-esteem and general wellbeing, and directly through support provision [6,26]. Friends become primary confidants for many individuals who self-harm, as particularly evidenced in qualitative studies [27–29]. Understanding, caring and empathetic responses from friends are especially valued. However, the mechanisms through which friends' support translates into self-harm protection over time, and how this support is perceived, remain largely unexplored.

Considering lived experiences and perspectives through qualitative designs provides in-depth understanding of the social context around self-harm and potential mechanisms and conditions underlying friends' positive and negative influence. Importantly, self-harm triggers, motivations and consequences differ over time and self-harm stages (cf. IMV model [11]). Adopting a temporal approach offers a lens through which to understand the relationship between self-harm and friendships over time, from self-harm onset to maintenance and beyond.

With this in mind, in the present study we adopted a retrospective qualitative framework – through the use of an adapted version of the Card-sort Task for Self-harm (CaTS [30]). The CaTS is a validated tool consisting of a deck of cards representing events, thoughts, feelings and behaviours, that participants sort along a timeline to describe their self-harm experiences. The tool allows participants to reconstruct the causal sequence preceding and following a specific self-harm event. This tool and framework has been adapted in this study to be used as

an accessible visual tool to facilitate sensitive discussions in a scaffolded manner [31], and to consider the temporal dynamics of friendships and self-harm from a first-person perspective.

### 1.2. The present study

We adopted a retrospective temporal approach to explore experiences of friendships and self-harm during adolescence. Qualitative interviews, prompted by the Card-Sort Task About Friends (CaTS-AF), considered the role, influence and meaning of friendships over time. Research aims were to:

a) Describe participants' perceptions of the meaning, role, importance and impact of friends on experiences of self-harm over time;

b) Explore how friendships and self-harm evolved and co-evolved over time.

## 2. Methods

### 2.1. Sample and recruitment

Eleven Psychology undergraduate students (F=9; M=2; aged 18–20 M=19.09, SD=0.70) from a UK university took part in the present study. Purposeful sampling was conducted through a university-based research participation platform, as part of a larger qualitative investigation which took place between October 2021 and December 2021. For the overarching study, participants were considered eligible if they identified as having self-harmed or having a friend(s) who self-harmed, or both, during adolescence (in this study we specified the period between 14–17 years old, given evidence of this being a key period for self-harm onset and maintenance [32]). Some participants had and discussed both their own and their friend's self-harm, but the present study is restricted to participants who indicated that they preferred to primarily discuss their own experiences of self-harm (see Fig 1 for information on recruitment process). Participants were also required to have access to a phone/laptop for the interview.

The study was ethically approved by the University of Birmingham Science, Technology, Engineering and Mathematics Ethical Review Committee (ERN_19–1815) and conformed to BPS guidelines for conducting research with human participants [33].

### 2.2. Study materials

**2.2.1. Baseline questionnaire.** Participants completed a demographic questionnaire reporting age and gender, alongside a baseline screening question where they stated the experience they preferred to primarily talk about in the interview (i.e., their own or a friend's self-harm).

**2.2.2. *Card-sort Task for Self-Harm About Friends (CaTS- AF)*.** The card deck of the original 117-card version of the CaTS represents individual experiences of self-harm [30]. In the current study we modified this task to increase the degree of focus on psychosocial and contextual elements related to friendships, in accordance with the study aims. The process consisted of an initial literature search of relevant studies considering social and friendship dimensions of self-harm [34,35]. DB compiled an initial list of possible cards to be added which were compared against the original task. Most additions were based on varying the wording of original cards (e.g., to capture interpersonal factors, or to simplify or combine similar cards) while in some cases, cards were created following an iterative process whereby proposed statements were discussed, selected and refined through consensus within the research team. Wording of both new and modified cards was selected based on relevance to

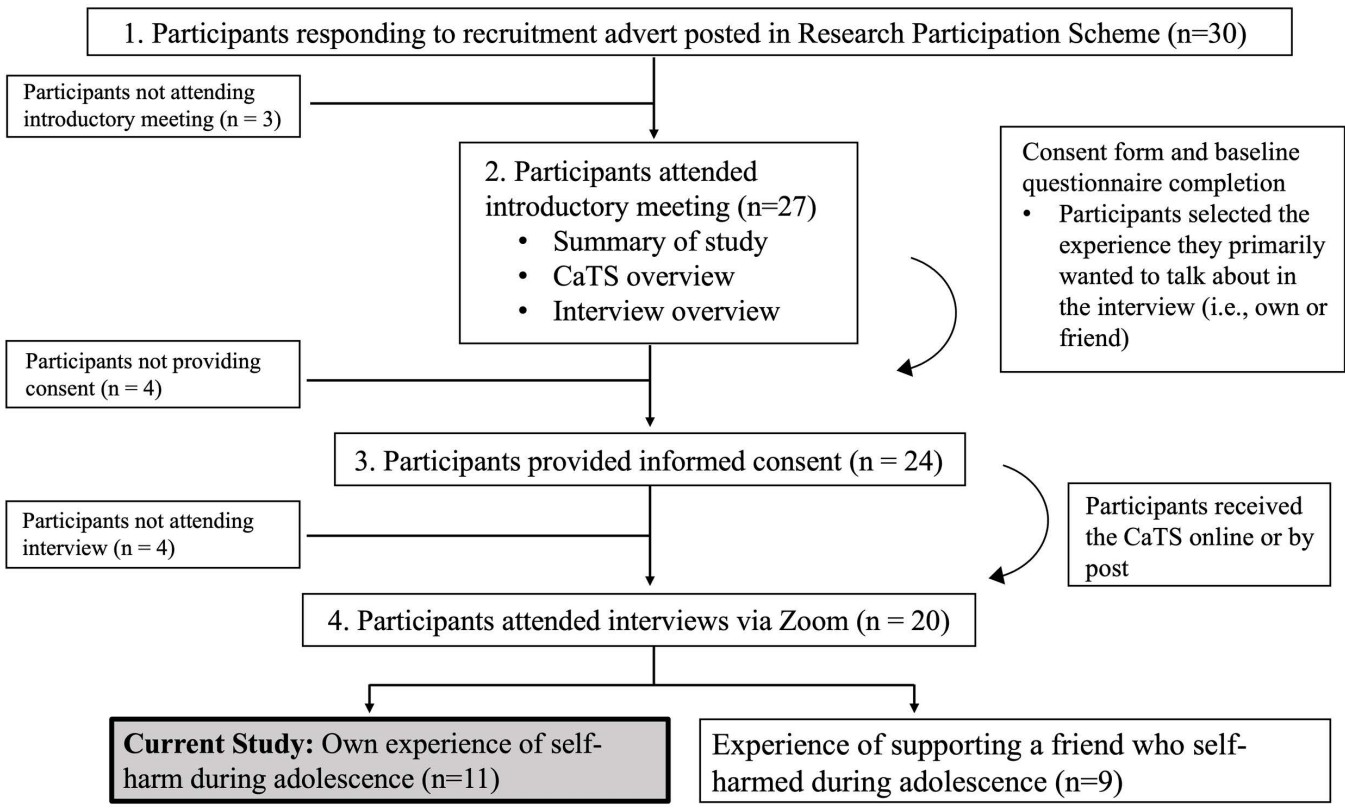

**Fig 1. Recruitment process for the present study.** Recruitment process for qualitative investigation of friendships and self-harm.

social dimensions noted in prior work. Examples include cards referencing social motivations and functions of self-harm (e.g., *I was trying to push people away; I got involved with a new group of friends who self-harmed; someone was dismissive*). This resulted in 18 new 'social' cards and 19 modified versions of original cards, resulting in the final 116-card Card-sort Task About Friends (CaTS-AF).

Participants were instructed to familiarise themselves with the CaTS-AF, pick cards that were relevant to their experiences (onset and most recent self-harm) and, if needed, write their own cards for experiences that were not already represented. Participants placed the selected cards along the timeline, and used stickers to identify cards they did not wish to discuss (Fig 2).

**2.2.3. Semi-structured interviews.** The interview schedule included questions centred around participants' self-harm experiences and trajectory, including: a) own understanding and experience of self-harm, b) friend relationships and the role of friends, c) characteristics and evolution of the friendship, and d) advice for supporting young people who self-harm (see S1 File for interview schedule). Two CaTS-AF arrangements corresponding to two key periods – self-harm onset and most recent self-harm – were used as prompts to discuss self-harm experiences over time. Using the CaTS-AF as a prompt intended to facilitate sensitive discussions in scaffolded manner, consistent with previous studies [24,31].

**2.2.4. Visual analogue scale (VAS).** Participants verbally rated their mood before and after the interview on a three-point VAS (Bad, Neutral, Good). Information from the VAS was not collected, instead it was used by the experimenter to identify and act on potential distress. A similar approach was used in previous studies [30].

| 6 Months Before | 1 Month Before | 1 Week Before | 1 Day Before | 1 Hour Before | I self-harmed | Afterwards |
|---|---|---|---|---|---|---|
| My friend had a problem at school | I received no support from caregivers | I wanted to push people away ● | I had an ✖ argument with my parent/ caregiver | I phoned a helpline which did not help | I did it on impulse without planning | I felt worse after self-harm |
| My friend was bullied/ cyberbullied ✖ | I felt worthless | I isolated myself from others | I was angry at someone | I felt there was no one to turn to for help | I had access to means to hurt myself | A + E staff were friendly and understanding |
| I was rejected by my friends/ peers ● | I felt sad and depressed | Someone suggested me to self-harm ● | I felt like a burden on people | I was very agitated and restless | I felt numb | I received therapy which helped |
| I felt rejected | I got involved with a new group of friends who self-harmed ● | | I talked to a caregiver which did not help | | I felt exhausted | I talked to a friend which helped |
| I felt like I did not belong | I discussed self-harm with my friends ● | | No one listened to me or took me seriously | | | I felt accepted/ I felt I belonged ● |
| I struggled with my sexuality ✎ | | | | | | I felt I could change for the better in the future |

**Fig 2. Sample CaTS-AF.** Sample CaTS-AF on Mural. Card colour denotes the category of each card, e.g., thoughts, feelings, events, behaviours, and support/ services. Originally blank cards on which participants wrote their own words are represented with a pen symbol. Crosses specify experiences the participant indicated they did not wish to talk about. Red dots denote examples of novel social cards developed by the research team for the present study and added to the original CaTS.

## 2.3. Procedure

The present study and interviews were conducted between October 2021 and December 2021. The study was conducted online due to face-to-face contact restrictions. Participants took part in an initial 1:1 Zoom session with the primary researcher to discuss the aims and conduct of the study and to familiarise themselves with the sensitive content. If comfortable and eligible to take part, participants completed a consent form and the baseline demographic questionnaire on Qualtrics and were sent the CaTS-AF before the next session. Depending on their preference, participants were given access to the CaTS-AF through a link to an online visual collaborative space (Mural), or received a paper copy by post. Consenting participants were then invited to a second and final Zoom session, in which the interview took place. Participants were instructed to familiarise themselves with the CaTS-AF, pick cards that were relevant to their experiences (onset and most recent self-harm) and, if desired, write their own cards for experiences that were not already represented. Participants placed the selected cards along the timeline, and used stickers to identify cards they did not wish to discuss (Fig 2).

Following verbal affirmation of consent, participants took part in the interview for which two CaTS-AF were separately arranged: a) participants arranged the CaTS-AF at the beginning of the interview to represent early self-harm experiences, i.e., self-harm onset, and b) participants arranged the CaTS-AF a second time, towards the end of the interview, to represent more recent experiences, i.e., recent self-harm. In both cases, participants either referred to a concrete memorable instance of self-harm or they described a range of experiences.

Both arrangements were used as prompts in the semi-structured interview which was audio-recorded using encrypted electronic and physical audio-recording devices. Audio-recordings were transferred to a secure University server, while interview transcripts were stored in a locked cabinet in the University (for paper copies) or in an encrypted file in a password-protected folder on a secure laptop (for electronic copies). Recordings were deleted when transcription was complete. Interviews lasted between 45 minutes and 1 hour and 30 minutes.

Research suggests that when handled with care and appropriate concurrent and follow-up support, online interviews may be as beneficial as in-person interviews, and may confer additional agency to participants [36]. In order to maintain ethical standards in an online setting, several steps were taken. First, DB checked-in with participants at various stages of the interview to ask them whether they felt comfortable to continue with the interview, whether they wanted to take a break or if they wanted to stop the interview altogether. This was typically done following CaTS completion and before/after specific questions. No participant chose to take a break or stop the interview, rather all stated they wished to continue. Second, after the interview, all participants were debriefed and signposted to sources of support (e.g., university counselling services, helplines) and DB engaged in informal conversation prior to closing the Zoom session to ensure participants were able to de-escalate from the potentially emotionally difficult conversation. Overall, no major distress was noted during the interview. Finally, participants were contacted again after two weeks to share thoughts or comments regarding their experiences of the study. Participants who responded to the email suggested not being negatively affected by taking part, but rather they felt that their participation allowed them to reflect on their experiences. This is in line with prior findings that asking about self-harm in research studies does not necessarily impact participants negatively and indeed may have relatively positive and/or perceived therapeutic benefits [37]. Follow-up comments were not included in data analysis.

## 2.4. Epistemological position

The philosophical framework adopted in the present research is pragmatism. This framework does not adopt any strict ontological or epistemological positionings. Instead, it assumes the existence of both individual and socially shared realities [38]. For that reason, this approach emphasises flexibility of methods, which are chosen based on their usefulness for exploring the phenomenon of interest. Given the focus on lived experience, we opted to utilise a qualitative approach enabling the identification of common contextual factors across participants, whilst also exploring how individuals subjectively interpret and make sense of their own individual experiences.

The first author, a female researcher undertaking a psychology doctorate research degree, undertook the interview and data analysis processes. The second author, a psychology professor with expertise in self-harm research, guided and assisted data analysis.

## 2.5. Data analysis

Data were transcribed verbatim by the first author. Eleven transcripts were included in the analysis, corresponding to the experiences of individuals who self-harmed in adolescence. We analysed the data using Reflexive Thematic Analysis (RTA) in NVivo v12 (QSR International Pty Ltd. Version 12 2018) following Braun & Clarke's guidelines [39]. RTA is a primarily interpretative and subjective activity occurring at the intersection between researcher's own positioning and subjectivity, the data and wider context [40].

The data analysis process started with an initial familiarisation with the data, which consisted of reading and re-reading all transcripts. Upon this, the first author line-by-line coded

all relevant transcripts, starting with simple descriptive codes which were later classified into broader and more abstract codes and categories. The analysis then focused on social and relational aspects of the experience, specifically friendships and peer relationships, in line with the research questions. At this stage, we developed and applied interpretative codes to the transcripts, these informing initial themes. Discussions with the research team, through a weekly peer-review process conducted with ET, and consultations of the literature helped make sense of and contextualise the findings. These processes helped refine and develop the final themes. A reflective diary of thoughts and observations was kept during interview and analysis, which informed theme-development and helped to reflect on the impact of the researcher during interviews, analysis and interpretation.

## 3. Findings

Interviews explored friendships, including their role and importance, their meaning, and their evolution over time among participants (aged 18–20) who self-harmed during adolescence (14–17 years old). Self-harm and friendships are complex, multifaceted phenomena and are experienced differently by each individual. Yet, three common aspects were identified across interviews, encapsulated within the following themes: a) *The role of friendships in self-harm progression,* which considers the variously positive, negative or limited roles played by friends in the experience, b) *The role of self-harm in friendship evolution,* which describes how self-harm influenced individuals' perceptions and experiences of friendships over time, and *c) The meaning of friendships in the context of adolescents' self-harm*, an overarching theme discussing the nature and meaning of friendships for young people who self-harmed (see Fig 3). The themes and corresponding quotes are presented herein.

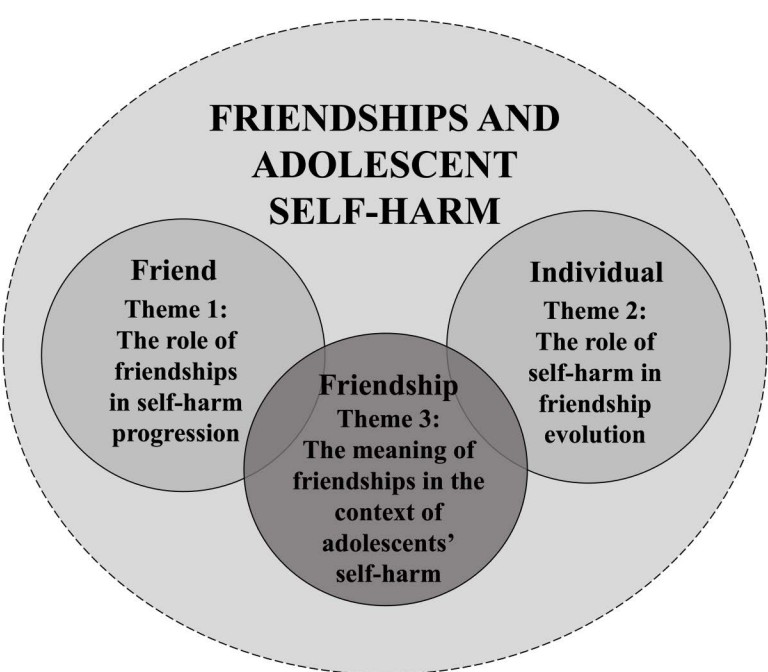

**Fig 3. Thematic map.**

### 3.1. Theme 1: The role of friendships in self-harm progression

This theme considers the role and influence of friends in shaping participants' self-harm. It describes different 'types' of friends perceived to hold distinct roles, including friends who had a negative influence, supportive friends, and friends who had a limited role in self-harm. Participants primarily discussed in-person same-aged school-based friendships, with few mentions of online or older friends. Therefore, the following results mainly reference the former group unless otherwise specified.

**3.1.1. Friends' influence on self-harm onset and maintenance.** A key theme mentioned in more than half of the interviews relates to friends' significant and adverse role in the experience of self-harm. Friends were considered significant contributors to participants' self-harm onset. Most commonly, having friends who self-harmed or engaged in other behaviours described as "self-destructive" was considered a starting point for participants' own self-harm. As one participant reflected:

> *"I think it all started when I got involved in, well they weren't a new group of friends, but the dynamic changed a bit… and they all started doing it (self-harming). And at the time, it was very kind of, (everyone) very much aside from me, did it"* (P7)

Friend's influence on participants' self-harm occurred in direct and indirect ways. For one participant, friends directly encouraged and ´bullied´ her into self-harming: "*They would say: "If you want to be a part of this friendship group and if you want us to stop harming you then you have to do this (self-harm)"* (P7). However, for the majority, friends did not actively encourage self-harm, but subtly introduced participants to it and normalised the behaviour as a possible coping strategy: *"She didn't encourage it or anything like that, but I think she made me realise that it was like another way to get rid of… to get out emotions"* (P6). A distinct pathway mentioned was the emotional impact of having and supporting friends who self-harmed: *"The idea that I had to look after her even though I didn't have to, kind of played into my mental health at the time"* (P4).

These experiences demonstrate that friends may be central in the initiation of self-harm. Across interviews, friends were also influential in self-harm maintenance, although this was less common. As self-harm became more habitual, the reasons underlying the behaviour became more relationship-oriented for several participants. Some of these reasons included arguments with friends: *"if they (friends) upset me or if they do something, it would really kind of upset me and gravitate me towards that (self-harm)"* (P2), friends *"freaking out"* or being unsupportive, and competitiveness or social comparison with friends and their self-harm whereby participants felt that their self-harm was not "severe enough". In some cases, friends would encourage each other to self-harm to cope. These examples illustrate how friendships may create a difficult interpersonal environment, maintaining and in some cases exacerbating self-harm:

> *"I think like a few months after I told one of my friends who also had self-harmed […] The first question she asked was: "How bad was it?". So I felt like that was a bit… I didn't know what to say cause then I just felt like I wasn't doing it right. Because I felt like she had done it worse and so she must have been feeling worse than me. […] It wasn't bad enough like I hadn't validated my feelings"* (P6)

> *"We fed each other unhealthy habits, because we only saw that (self-harm) as like a normal coping mechanism when it wasn't […] it's not a healthy way to do it. So we were close, we were really close, but I think maybe… a bit too close"* (P4)

Overall, whilst friends seemed to negatively contribute to adolescents' self-harm, their influence alone did not account for the behaviour, but rather it provided additional strain to already present vulnerabilities. One participant recalled not being impacted by their friends' self-harm until they started having their own struggles: *"It didn't make me think: "Oh I want to do that too", because I was fine at the time, like I wasn't doing bad mentally"* (P24).

**3.1.2. Friends' provision of support.** Most participants described having friends who provided support throughout the experience of self-harm: *"They are a good support system, friends that I can always go to that will be there"* (P10).

A few participants confided in a close friend prior to self-harming for the first time: *"I was trying to avoid doing it and I think me talking to her was kind of… a cry out for help, hopefully she would say something that would change my mind"* (P19). However, the majority suggested that friends were the first to learn about self-harm, once it had already happened. Participants stated that they either opened up to friends or their friends recognised noticeable changes in participants' behaviour and further asked about it:

> *"I think he sensed, before I did, that something was wrong. 'Cause he knew, 'cause obviously he knew me quite well […] He sort of did the detective work and found out: "oh yeah, you're not doing great" and then I sort of spilled it out to him"* (P9)

Across these instances, it appears participants were open to discuss self-harm and its underlying causes with friends. In response, friends' attitudes and reactions in most interviews were characterised as positive and supportive, including distraction and practical, informational and emotional support. Some friends also actively encouraged formal help and convinced participants to seek professional support: *"I kind of had a few friends that knew about it, that I talked, would talk to about it that kind of encouraged me to get some more (help)"* (P24):

> *"They were very open to have discussions, in the sense that they would always offer themselves as a source of support […] That made me feel comfortable"* (P19)

Overall, the importance and positive role of supportive friends was recognised and valued by participants during the experience and in the present day.

**3.1.3. Friends played no role or a limited role in self-harm.** In some instances, participants alluded to the fact that friends did not necessarily play a significant role in their self-harm. One participant explicitly mentioned not involving their friends and never sharing their self-harm with them, these having no role in their experience. Her case, whilst uncommon, reflects a view shared by several participants who reflected on the limitations of friend's role given their position as same-aged peers:

> *"They weren't really aware of anything, that was going on with me. And that's the way I wanted it and then, so… again there was nothing they could really have said or done that would have changed the way I was feeling"* (P11)

> *"(friends) had very limited knowledge about stuff, so their sort of support was very day-to-day stuff but then… my teacher was obviously a lot older and had been through struggle in her own life, so she helped me guide in a more sort of mid-term and long-term planning"* (P9)

Similarly, as participants grew older and learned additional coping strategies, they felt less need to share their self-harm and rely on friends for support: *"It (self-harm) is not something I have to share with people if it does happen"* (P24).

*"But I feel like now I'm quite… I'm a bit more, I'm a lot more resilient than I was, so I kind of prefer just dealing with it by myself until I really need to speak to my mate or my mum"* (P4)

The present theme suggests that friendships had an important role throughout the experience of self-harm, yet this appeared to change over time, where friends became less influential over time both as stressors and sources of support.

### 3.2. Theme 2: The role of self-harm in friendship evolution

Self-harm is recognised as a primarily individual experience, nevertheless it is situated within its social context. The present theme explores how self-harm shaped friendships over time.

**3.2.1. The isolation and loneliness paradox.** The period prior to and following self-harm was generally described as a period of extreme negative feelings which impacted participants' behaviours and social relationships. Negative feelings were often precursors to behavioural changes. Among these, one particular change recalled across all interviews was isolating from others and pushing others away. Several participants suggested isolation was related to feeling low and not having energy to be with others, not wanting to affect those around them or wanting to justify the way they were feeling. Also, isolation was in some cases a by-product of trying to avoid talking about self-harm or the issues underlying it: *"I didn't really feel that I could talk, because obviously the whole reason I was feeling like this was literally centred all around my sexuality"* (P24).

*"I've always isolated myself when I felt bad cause it was almost like, you don't want to affect other people […] Also, part of me thought If I give myself a reason to feel as bad as I do it will make sense"* (P20)

Some participants also suggested that their isolation could be perceived as them pushing others away. They explained that self-harm contributed to negative feelings and anger, which partly were targeted at others: *"I feel like my friends just didn't feel as close to me anymore and they didn't think that I wanted to be with them, because when I get sad or emotional I kind of turn it into anger"* (P4). Participants were aware of the repercussions of their behaviour in others: *"I wouldn't say I was explicitly trying to push people away, but I think that resulted in people gravitating away from me, because of that attitude"* (P2).

Paradoxically, most participants suggested that feeling a lack of connection to others, or that others did not care, was the primary motivation for isolating themselves and pushing others away. However, this consequently amplified unwanted feelings of loneliness: *"It's not like I wanted to be on my own, I just thought I was"* (P11):

*"Some of my friends noticed (isolation), but I didn't really take it seriously that they noticed. I think because I kind of managed to convince myself that nobody, like genuinely cared"* (P21)

Overall, negative feelings appeared to contribute to isolation which further reinforced adolescents' beliefs and distorted perceptions of themselves and others. Ultimately, this served to amplify their negative feelings in a cyclical manner. This highlights an important pathway towards self-harm maintenance and difficulties accessing support.

**3.2.2. Disclosure: Contradictions, expectations and help-seeking.** Feelings of loneliness and isolation are particularly relevant for understanding participants' attitudes towards disclosure and help-seeking.

First, most participants agreed that at the time they needed and wanted help, but as a product of their isolation, they felt help was not readily available or accessible: *"You kind of don't know who to go to or what to tell people […] But you also do really want the help"* (P19):

> *"Wanting to turn to someone for help and not being able to… made the feeling worse. I feel like when you are kind of in your own head and by yourself, that's when things can get really difficult because you don't have anyone to kind of challenge your drastic thoughts"* (P21)

Although participants wanted help, most expressed difficulty seeking help through hesitancy in trusting others given fear of extreme reactions, fear of upsetting friends and fear of being reported: *"I gave them (friends) enough information to fulfil their own need of wanting to know, but not too much that it would be reported"* (P2). In addition, some suggested struggling with explicitly asking for help and expected friends to notice signs and to initiate these difficult conversations: *"maybe if like a friend noticed or something, then maybe I would have been a bit more likely to say something"* (P11).

Altogether, despite young people's expressed need for support, these additional barriers may have precluded help for many participants.

**3.2.3. How self-harm shaped the course of friendships.** Participants' perceptions and experiences of friendship in the context of self-harm contributed to shaping these relationships over time, mainly in two opposite directions.

In a small number of cases, friendships broke down and ended: *"(friends) weren't really in my life anymore"* (P6). This was partly in relation to friends' negative and stigmatising reactions to self-harm, often described as unhelpful: *"(Friend) completely freaked out on me and that just stressed me out so much […] It just made me feel so ashamed and also made me feel like I was affecting her or something"* (P20).

For most, the experience mainly improved the quality of participants' relationships. Given the nature of the support exchanged and the level of intimacy created, some friendships grew closer *"I think we're also a lot closer as well and we were all sharing stuff like that"* (P10). As explained by one participant:

> *"I guess I'm closer with my best friend because of it, because, like she's really seen me at like really dark points. So I think someone like that, you can't really ever let go of someone that's really helped you through your really bad times"* (P24)

Overall, self-harm played a role and influenced friendships in various ways. Not only did it contribute to participants' perceptions of their friends, but also their behaviours towards them, ultimately shaping the course of these relationships over time.

## 3.3. Theme 3: The meaning of friendships in the context of adolescents' self-harm

The aforementioned themes reflect how self-harm both reaffirmed, challenged and shaped participants' notion of friendship: *"I think obviously in the past, my perception of what a friend is has been quite twisted because of my past experiences, whereas as I've grown up I kind of know what a real friend is"* (P7). The following subthemes consider participants' understandings and meanings of friendship in the context of self-harm.

**3.3.1. The value of relatability and mutual understanding.** All participants reflected on the importance of relatability and mutual understanding within their friendships. That is, the ability to relate to friends, and for friends to be able to relate to and understand participants' experiences in return.

Participants described self-harm as a frequently misunderstood behaviour. For this reason, they placed special value and trust in friendships where they felt understood: *"I talked to one of my close friends about it […] it was quite nice to have someone there who understood what I was going through"* (P10). As one participant pointed out, rather than closeness, relatability became a more important feature for their friendship:

> *"The first friend I spoke to, we were close but we weren't best friends. But I think the thing was, I knew she'd understand […] She would understand in any way even if she couldn't personally understand, I knew she struggled with her own mental health and she'd told me stuff before. So that was our friendship"* (P20)

Conversely, feeling misunderstood and misrepresented was one of the main concerns participants expressed: *"I think I was very hesitant in telling certain people stuff and I think I was just trying to be careful […] saying things around certain people just because, I wasn't sure of how they'd take it"* (P19).

Overall, participants' impressions of friends' ability to relate to them or effort to understand them influenced their level of trust and the extent to which they relied on them for support.

**3.3.2. Friendships are reciprocal and balanced.** An additional feature encapsulated within the previous subtheme is that of reciprocity and balanced friendships, where both are equals and are able to reciprocate support.

First, participants' worries about being perceived differently were consistent with their wish to be treated as an equal, despite their circumstances: *"I didn't want to be treated with sympathy, or seen as vulnerable, I wanted to still be treated like everyone else"* (P4). Therefore, participants considered that given a certain level of support was sought from friends in the context of self-harm, it was important for them to be able to reciprocate it: *"I'd speak to her quite a lot, but I think she spoke to me quite a lot as well"* (P20).

Reciprocity of support was re-negotiated throughout the experience in order to find a new balance within the friendship. In contrast, the absence of reciprocity through one-sided and non-reciprocated friendships negatively impacted participants: *"I felt rejected […] and I was also quite angry because I was there for them when they were going through what they went through earlier and I felt a bit like, I wasn't quite requited"* (P6):

> *"There was a lot of using me, kind of taking advantage of how nice I was, 'cause I would end up being kind of the therapist friend, but I wouldn't get that in return. Because when it came to my problems, I'd just kind of get dismissed"* (P21)

Overall, participants wanted friends to treat them as equals. They also reflected on the importance of reciprocity, especially in terms of mutual support.

**3.3.3. Respecting boundaries within friendships.** Despite participants' expectations of support from friends, over time there was growing awareness that this could not always be the case and that friendships should have boundaries.

On one hand, the importance of respecting friends' boundaries was particularly discussed by participants who had also supported a friend who self-harmed, i.e., making them both a recipient and provider of support. Through their personal experiences, participants became mindful that support provision in the context of self-harm may negatively impact supporters and 'trigger' others: *"Because I know how much it impacted me… I didn't want someone else to go through that"* (P24). Therefore, they suggested it was important to respect their friends' boundaries and capabilities:

*"I think you realise as you get a bit older that you can't keep going to the same person because they have their own stuff to deal with […] You do need to kind of spread the load a little bit or... if you keep going to the same person, you're emotionally draining to them"* (P20)

*"People that I knew, maybe weren't doing great themselves, I wouldn't want to go to them […] I'm very aware of that, because I don't think, I think it's a big thing to be able to take on someone else's stuff when you have your own stuff going on"* (P24)

On the other hand, several participants expressed appreciating friends' sensitivity and respect for participants' space, boundaries, and level of comfort when discussing certain topics: *"I feel like when speaking to them about it, they could see that I wasn't… comfortable enough talking about it so they kind of left it but made sure that I knew they were there for me"* (P19).

Overall, various aspects of friendships became gradually more important throughout the experience of self-harm, these all reflecting some form of reciprocation.

## 4. Discussion

In the present retrospective qualitative study we explored experiences of friendships and self-harm in a sample of psychology undergraduate students from a UK university who self-harmed during adolescence. Semi-structured interviews were prompted by the Card-sort Task for Self-Harm About Friends (CaTS-AF) and analysed using Reflexive Thematic Analysis. Three themes were developed: a) *The role of friendships in self-harm progression*, b) *The role of self-harm in friendship evolution,* and c) *The meaning of friendships in the context of adolescents' self-harm*. The first two themes represent the influence of friendships and self-harm on one another – from each corresponding direction – emphasising the interdependent nature of the two constructs. The third theme provides an overarching commentary on the nature of friendship and its acquired meaning throughout the experience.

### 4.1. Adolescent friendships and their role in self-harm

Friendships are an important part of young people's lives. Adolescent friendships play a major role in distress and risk-taking behaviours but are also recognised as important sources of support [41,42]. Self-harm onset typically occurs during this period, suggesting adolescence may provide a unique developmental context in which these two phenomena become more salient and intertwined.

The present findings underscore traditional views of adolescent friends as sources of both positive and negative influence on self-harm. First, the negative role and influence of friends' self-harm was mentioned across the majority of interviews, in line with quantitative evidence suggesting that exposure to friends' self-harm increases the risk of self-harm [18,22]. The current study highlights possible mechanisms underlying friends' influence, including direct influence (e.g., coercing, suggesting, imitating, normalising) and indirect influence (e.g., through increased distress). At the same time, the balance of quantitative and qualitative evidence unequivocally agrees that friends are also primary sources of support for young people who self-harm [25,43]. In this study, participants reflected on the positive contribution of friends' support in reducing self-harm, increasing positive feelings and encouraging support-seeking. Similar to previous qualitative studies, participants in our sample agreed that helpful responses from friends included empathetic understanding, caring and non-stigmatising responses alongside distinct types of support offered, including emotional, practical and physical support [28,29]. This highlights the value and importance placed on friendships.

The qualitative temporal approach we adopted in the present study provides novel perspectives on the interdependence of friendship and self-harm evolution. Particularly, we highlight

the complex role of friend support, situated at the intersection between positive and negative influence. Support can contribute to both positive and negative outcomes for the receiver, depending on its nature (empathetic understanding vs stigmatising responses) and how it is perceived (helpful vs unhelpful). Equally, support provision may also have positive and negative consequences for the provider, including increased mental health awareness and friendship closeness, but also negative feelings, stress and increased risk of self-harm. The responsibility of supporting a friend through self-harm appeared to interact with participants' own struggles, in some cases precipitating self-harm (cf. [44,45]). This emphasises the interrelatedness of individual and friend characteristics, these evolving over time and in response to one another.

Finally, it is also worth noting that a minority of participants never disclosed their self-harm to or accessed support from any friends. In effect, this is not an unusual occurrence since a substantial proportion of adolescents who self-harm (40%) report not accessing any type of support [46]. Even the decision of not involving friends is significant and can provide valuable insight on individuals' experiences. It also highlights the need to consider the broader social ecology, in which friendship remains an important but by no means the only aspect.

## 4.2. Impact of the self-harm experience on friendships

Findings suggest that self-harm also plays a role in and influences friendships, including by distorting perceptions of available support. To illustrate this, we reflect on findings around help-seeking and isolation. Help-seeking is dependent on having or perceiving that support is available [47], as well as on individual differences in need for and predisposition to seek help [48,49]. Across interviews participants expressed the need for support, yet also displayed struggles in explicitly seeking help and accessing appropriate support [46]. This can be attributed to reasons emerging from the individual, their social environment and the interaction between the two.

First, feelings of shame, burdensomeness and thwarted belongingness are characteristically experienced in the context of self-harm [11,50,51].As observed in the present study, these feelings may influence individuals' predisposition to seek help and their perceptions of the availability of support. Interestingly, these same feelings also contributed to social withdrawal and isolation. In some cases, isolation represented a non-spoken language to signal distress and seek help, potentially given adolescents' inability to recognise and/or verbalise their complex emotions, alongside shame and fear for others' reactions [52]. However, when this signal is not responded to, this may further reinforce negative feelings and beliefs, reducing opportunities for others to identify and respond to distress and self-harm, ultimately precluding support.

This important insight may have practical implications for the identification of at-risk individuals. Importantly, it highlights that friends may inhabit a unique position to recognise and respond to adolescents' signals of distress and self-harm, given their status as same-aged peers. Friends' shared developmental experiences may account for early recognition of self-harm, alongside understanding attitudes and responses, such as empathetic concern [24]. This unique expertise and role should however be apprehended alongside awareness of the potential for distress to be experienced among young supporters [53]. Therefore, educational and clinical professionals working with adolescents who self-harm should additionally identify and extend support to those in their close social network, specifically friends.

## 4.3. The acquired meaning of friendships in the context of self-harm

Finally, findings from the present study suggest that friendships acquire specific meanings, dynamics and expectations within the context of self-harm, whereby traditionally valued

friendship characteristics may become more prominent in this context while others may be perceived more negatively.

For example, the presence of trust, mutual understanding and shared experiences has been found to relate to help-seeking, the acceptance of support and perceptions of helpful responding [29,54,55]. Within the present sample, shared understanding through similar experiences – including self-harm and other mental health struggles – contributed to the perception of less stigmatising and more supportive responses from friends, further encouraging disclosure (cf. [28]). In other accounts, the impact of lack of understanding from friends and the negative consequences and feelings resulting from it were noted. This mirrors research in other stigmatised populations, e.g., sexual and gender minorities [56,57]. Understanding how adolescents ascertain perceived similarities within friendships should be explored further, given their important implications for help-seeking, mutual support and friendship quality.

Instead, certain friends' caring responses were perceived unfavourably by participants. For example, friends treating participants differently or friends disclosing participants' self-harm to others were at odds with their expectations of loyalty, trust and secrecy in regards to self-harm (cf. [28]). This reflects conflict within the expectations of friendships and the competing interests emerging within a friendship in the context of self-harm. Importantly, these positive and negative experiences with friends may influence perceptions of friendship, ultimately shaping these relationships and the outcomes experienced by young people and friends.

### 4.4. The temporal transition of the individual, the friend and the friendship

The findings discussed highlight that both friendships and self-harm are complex, dynamic, evolving experiences. Adopting a temporal approach, particularly through the use of the Card Sort-Task for Self-Harm About Friends (CaTS-AF), enabled exploration of the temporal pathways leading to and following self-harm and the changing role of friends [30,31]. For instance, we note that whereas friends in our study had a prevalent negative role during self-harm onset, they were primary sources of support following disclosure, and subsequently their role seemed to diminish as individuals entered young adulthood, consistent with extant evidence [22,23,28]. In the current findings these roles were variously a) present in different friends, b) co-existent within the same friend or c) evolved from positive to negative, or vice versa, within the same friendship, highlighting the multiplicity of friendship roles and experiences. As such we recommend future studies continue to adopt an explicitly temporal approach to understand adolescents' needs and social ecosystems at different stages of self-harm.

### 4.5. Reflexivity and researcher's position

The use of qualitative methods inherently assumes that research is contextual and subjective [58,59]. This subjectivity and the researchers' own positions, characteristics, experiences and their roles in shaping the research, should be reflexively considered [58,60].

Particularly, my position as a doctoral student shaped the research in various ways across the different stages of the research process. First, being a student implied initial limited knowledge around the topic and how to best approach it. This informed the decision to allow participants to self-identify and describe their own self-harm and friendship experiences, without a priori constraints. This approach may have facilitated interviews by allowing participants to feel in control of the interview process and may have helped validate their experiences. Equally, this may have made the interview process more challenging in some cases due to the broad and non-restrictive nature of the questions. However, the use of the CaTS scaffolded the process for participants.

Additionally, similarities between myself and participants also shaped the research process and interpretation of findings. On one hand, being relatively similar in age and gender to most participants appeared to facilitate certain conversations and reduce power dynamics. In turn, it promoted relatability and shared understanding of the context in which certain self-harm experiences occurred (e.g., Tumblr, online friendships). On the other hand, similarities in age and educational background (i.e., psychology) may have shaped conversations in a particular way. During interviews, some participants appeared to use more clinical language and reasoning (e.g., conversations about warning signs/ symptoms of certain mental health disorders and how they manifested in themselves or others). Whilst this helped create mutual understanding, it also appeared to lead to more neutral and objective descriptions and sense-making, potentially detracting from more experiential aspects of participants' accounts of their subjective experiences.

### 4.6. Limitations and suggestions for future research

The present findings have certain limitations. First, despite the temporal view we adopted, we explored only a limited period, from early/mid-adolescence to young adulthood. Recent research highlights the potential for earlier self-harm onset (i.e., during childhood) and maintenance beyond young adulthood [61]. Nonetheless, our adolescent/young adult focus provides in-depth insight on a period of key relevance, given typical ages of self-harm onset, peak and decline [62].

Second, the size and characteristics of our sample may have implications for the findings presented. The sample size was determined based on the overall research aim of providing an in-depth exploration of individuals' experiences and sense-making. For this aim and when using an RTA approach, smaller samples of 10–20 participants are typically recommended [63]. Despite the richness of the data captured in this study, it is important to reflect on the limitations of our purposeful sampling approach in sufficiently capturing participants' experiences. Specifically, the sample consisted of undergraduate psychology students who may be a relatively homogeneous group with similar demographic and sociocultural characteristics and whose unique institutional context may have introduced potential population bias. As such and as mentioned in prior sections, the self-harm experiences of this sample may have certain similarities in trajectory and processes of sense-making. Therefore, it is plausible that certain self-harm experiences – where participants displayed balanced evaluations of their current situation in relation to both self-harm and friendships – were over-represented in our findings. Potentially, individuals with more difficult or unresolved experiences in the past and present chose not to participate, with implications for interpreting the current findings.

Representation should also be considered for two particular population groups present in the study, albeit not fully explored, namely males and sexual minorities. These two groups are documented to present a) more severe manifestations of self-harm, such as increased risk of suicide, b) specific norms around emotional needs and social support (e.g., masculinity) and c) increased barriers for disclosure and help-seeking [64,65]. A number of these characteristics were described as playing an important role in participants' experiences, and should therefore be explored further.

### 5. Conclusions

In the present retrospective qualitative study we explored the experiences of friendships and self-harm in eleven individuals who self-harmed during adolescence. The present findings underline the role of friends as sources of risk, through their own self-harm behaviour, as well as sources of protection through empathetic understanding and support. Rather than being separate, these two roles often intersected. Second, isolation consistently emerged across all

participants as a behavioural manifestation of distress, impacting friendships and contributing to self-harm maintenance for example by precluding support-seeking. Adolescent friends emerge as a group uniquely suited to recognise this subtle interpersonal signal of distress. Yet, the potential risk of self-harm and adverse outcomes when adopting a supporter role should not be overlooked.

Adding nuance to this evidence, findings suggest complex and dynamic interactions of protective and risk processes occurring at the individual, friend and friendship level. It is essential to identify significant peer relationships within the social ecosystems of young people who self-harm. The unique role and position of friends warrants their consideration when developing awareness and educational prevention strategies and clinical interventions, which acknowledge the important role that friendships have in understanding self-harm.

## Supporting information

**S1 File. Interview schedule.** Summary of interview schedule.
(PDF)

## Acknowledgments

Townsend acknowledges the support of the UK Research and Innovation (UKRI) Digital Youth Programme award (MRC project reference: MR/W002450/1) which was part of the AHRC/ESRC/MRC Adolescence, Mental Health and the Developing Mind programme. We would like to thank the young people who generously gave their time for this study.

## Author contributions

**Conceptualization:** Delfina Bilello, Ellen Townsend, Matthew R. Broome, Stephanie Burnett Heyes.

**Data curation:** Delfina Bilello.

**Formal analysis:** Delfina Bilello, Ellen Townsend.

**Funding acquisition:** Ellen Townsend, Matthew R. Broome, Stephanie Burnett Heyes.

**Investigation:** Delfina Bilello, Stephanie Burnett Heyes.

**Methodology:** Delfina Bilello, Ellen Townsend.

**Project administration:** Delfina Bilello, Stephanie Burnett Heyes.

**Resources:** Ellen Townsend, Matthew R. Broome, Stephanie Burnett Heyes.

**Supervision:** Ellen Townsend, Matthew R. Broome, Stephanie Burnett Heyes.

**Validation:** Ellen Townsend, Matthew R. Broome, Stephanie Burnett Heyes.

**Visualization:** Delfina Bilello.

**Writing – original draft:** Delfina Bilello.

**Writing – review & editing:** Delfina Bilello, Ellen Townsend, Matthew R. Broome, Stephanie Burnett Heyes.

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
