## [Decision Letter · Decision Letter 0]

29 Oct 2024

PONE-D-24-36301“People in my life just play different roles” A retrospective qualitative study of friendships among young adults who self-harmed during adolescencePLOS ONE

Dear Dr. Bilello,

Thank you very much for submitting your manuscript to PLOS ONE. After careful consideration, we feel that it has merit but does not fully meet PLOS ONE’s publication criteria as it currently stands. Therefore, we invite you to submit a revised version of the manuscript that addresses the points raised during the review process. While there is much to admire in your paper, in particular, having read your paper and the referees' comments, I would like you to address carefully the comments of both referees, who suggest that your manuscript needs to discuss better the following points:

1) Greater consistency and clarity when discussing the types of friends involved in the study. 

2) Inconsistency in the terminology used to describe the participants' developmental stages throughout the manuscript. 

3) Think carefully about potential limitations to your study.

4) Further reflection on how the researcher’s position as a student may have influenced their interactions with participants, who were also students.   Please submit your revised manuscript by Dec 13 2024 11:59PM. If you will need more time than this to complete your revisions, please reply to this message or contact the journal office at plosone@plos.org . Please include the following items when submitting your revised manuscript:

We look forward very much to receiving your revised manuscript and thank you for submitting your research to PLOS ONE.

With kind regards,

Professor Sriya Iyer

Academic Editor

PLOS ONE

Reviewers' comments:

Reviewer's Responses to Questions

**Comments to the Author**

1. Is the manuscript technically sound, and do the data support the conclusions?

Reviewer #1: Yes

Reviewer #2: Yes

2. Has the statistical analysis been performed appropriately and rigorously?

Reviewer #1: N/A

Reviewer #2: N/A

3. Have the authors made all data underlying the findings in their manuscript fully available?

Reviewer #1: No

Reviewer #2: Yes

4. Is the manuscript presented in an intelligible fashion and written in standard English?

Reviewer #1: Yes

Reviewer #2: Yes

5. Review Comments to the Author

Reviewer #1: OVERALL SUMMARY

Thank you for the opportunity to peer-review this important manuscript on the impact of friendships among young adults who self-harmed during adolescence. The paper addresses an underexplored topic in self-harm research, offering valuable insights into the interrelationship between friendships and self-harm, and the complexities involved.

The study’s findings emphasise that friendships not only influence self-harm behaviours but are also shaped by them, which is an important contribution to the field. The use of semi-structured interviews within a retrospective qualitative design, alongside a modified CaTS sort task and reflexive thematic analysis, constitutes a strong and appropriate methodology for capturing the subjective experiences of participants. It was particularly commendable to see the focus on both the onset and maintenance of self-harm in relation to friendships. The CaTS-AF task was used effectively to capture these temporal aspects, and the researchers made full use of this tool to enhance their analysis. Overall, the study identifies relevant and significant themes, and the interpretation of the findings is well-grounded, offering fresh perspectives on the connection between adolescent friendships and self-harm.

MAJOR CONCERNS

However, I have a few concerns that might be considered more “major”.

1) There needs to be greater consistency and clarity when discussing the types of friends involved in the study. It’s not always clear whether the focus is on same-age friends, online friends, or offline friends, as these distinctions are not consistently made in the methods or results. For example, the method section on line 136 notes that the interview schedule included questions about participants’ “offline and online friend relationships,” but in the results, there isn’t always clarification about which type of friend is being referred to, assuming participants specified this at all. At times, same-age peers are mentioned, but it’s not explicitly outlined whether this is the primary type of friend captured throughout the study. If the intention is to refer to "friends in general" unless otherwise specified, this should be made clear in the results, particularly since the majority of interviewees appear to discuss same-age friends. For example, in the discussion, the phrase "given their status as same-aged peers" (line 529) is used, but it’s unclear if participants consistently referred to same-aged peers throughout the study. This ambiguity could be addressed by more clearly specifying in the methods and results which types of friends—whether peers, online friends, offline friends, or same-age friends—are being discussed. If there were different types, it might be important to reflect on this in the discussion.

2) There is some inconsistency in the terminology used to describe the participants' developmental stages throughout the manuscript. Terms such as "adolescent" "young adult" "young person” are used. Given that adolescence is now often considered to extend up to age 24 (e.g., Sawyer et al. 2018 “the age of adolescence”), university students can still technically fall within this period. And given that the mean age of participants was 19 (with a range of 18 – 20) it might help to clarify that the developmental stage being discussed is still adolescence, but with self-harm recollected from early adolescence. This would ensure alignment with the current understanding of adolescence as extending into the mid-20s.

3) The limitations section could be expanded to address some important aspects that are currently missing. At present, the section is brief, only spanning two paragraphs, but there are several additional limitations that could be considered. For instance, the fact that the sample consists entirely of university students introduces an undergraduate population bias. The unique cultural and institutional context of the university may shape the participants' experiences and responses, and this should be acknowledged. While qualitative research is not focused on large sample sizes, but rather on the concept of data "saturation," it could still be valuable to address this explicitly in the limitations section. A brief comment on how saturation was approached or achieved would strengthen the methodological transparency of the study.

4) Lastly, it would be useful to include further reflection on how the researcher’s position as a student may have influenced their interactions with participants, who were also students. This shared status could have affected the rapport between the researcher and participants, as well as the dynamics of the interviews. For instance, participants might have felt more comfortable or, conversely, may have withheld certain views based on perceived similarities or differences. On line 200, it is noted that a reflective diary was kept to account for the researcher’s impact during interviews and the analysis process, which is an important practice. However, it would be valuable to expand on this reflection in the discussion section. Even a brief mention of the specific insights or challenges arising from the researcher’s reflexive process could provide important context for interpreting the findings. If word count is a concern, this discussion could be summarised in one or two sentences within the main text, with more detailed reflections placed in the supplementary materials. A comment on reflexivity in the discussion, whether brief or expanded, would strengthen the study’s rigor by demonstrating an awareness of how the researcher’s positionality may have shaped the research process.

MINOR SUGGESTIONS

Additionally, below are some minor suggestions specific to different sections of the manuscript, that may help to improve it further.

TITLE/ABSTRACT

• Title could better reflect the study by saying “role of friendships” e.g., [“People in my life just play different roles” A retrospective qualitative study of the role of friendships among young adults who self-harmed during adolescence”]

• Abstract – lines 9/10 – might be better to have the (M=19.09; SD= 0.70; M=2, F=9) after the phrase young adults – because I read that as though the age they self-harmed was a mean age of 19.09 rather than that being the age of the sample.

• Line 15 , could say “but they acquire” rather than “these acquire”.

INTRODUCTION

• Line 56– maybe add in what friendships are defined as in the first place e.g., same age friends? Perhaps just outlining exactly what is meant by friendship in this context (this links in with the more ‘major’ concern comment 1 above).

• On line 97, the research aim states “Describe participants’ perceptions of the role, importance and impact of friends on experiences of self-harm over time”. However, the study is also focused on exploring what the “meaning” of friendship in relation to self-harm is too, so that could be added/reflected in the aims. Also a suggestion - wonder if the aim stating “explore how friendships… were shaped by self-harm” the phrasing might suggest an assumption that self-harm shaped friendships, rather than an open exploration of all possible dynamics between self-harm and friendships i.e., implies that you are already expecting to find this. I appreciate that data has already been collected and analysed and this may have been your aim, but if your aim was intended to be more neutral and it is not too late to refine the wording for clarity and neutrality - instead of “how friendships were shaped by self-harm” - a more neutral phrasing could be “explore the relationship between self-harm and friendships” or “explore how self-harm and friendships interact”, which doesn’t assume a one-way effect.

METHODS

• Line 128 – 133 in the study materials section “Depending on their preference, participants were given access to the CaTS-AF through a link to an online visual collaborative space (Mural), or received a paper copy by post. Participants were instructed to familiarise themselves with the CaTS-AF, pick cards that were relevant to their experiences (onset and most recent self-harm) and, if needed, write their own cards for experiences that were not already represented. Participants placed the selected cards along the timeline, and used stickers to identify cards they did not wish to discuss (Figure 2)” might be better placed in the procedure section on/after line 153.

• Line 89 “here we adapted this tool and framework to the study of friendship and self-harm to consider the dynamics of friendships and self-harm from a first-person perspective” – not quite clear exactly how you adapted it – what was the process of adapting it? You mention on line 124 “adding 18 new social cards based on team discussions and relevant literature” but could maybe add a bit more about how these cards were derived/any things that were discussed but maybe did not go in there and if there was any kind of consensus approach to this.

• Line 169 “participants who responded emphasised positive aspects of participation (Dazzi et al., 2014)” – could state what they were/unpack this a bit more as it’s not quite clear what you mean. Also not sure if this citation should be here i.e., the reference is titled “Does asking about suicide and related behaviours induce suicidal ideation? What is the evidence?” - which may/or may not be relevant to this comment.

• Ethical considerations – The manuscript could benefit from more detail regarding how potential participant distress was handled, especially given the sensitive topic of self-harm. For example, it would be useful to clarify how the study followed up on participants who might have experienced distress during or after the interviews (e.g., line 145 mentions addressing potential distress). It would be helpful to specify what steps were taken to mitigate this, such as offering support or referrals, and whether any participants required follow-up. If relevant, including this in the results section would provide transparency about how such situations were managed

• It would be beneficial to provide examples of the specific questions or prompts used in the interviews, to give readers a clearer understanding of how data was collected. Additionally, please clarify how the interviews were handled after data collection, particularly in terms of data storage. For instance, were the recordings deleted after transcription, and if so, what were the procedures to ensure data security? It would also be helpful to mention whether the interview guide evolved during the study and, if so, how changes were made throughout the course of data collection.

RESULTS

• Line 337: I may be misunderstanding the concept, but I’m unclear about how the "notion of isolation appears as a paradox given that participants also reported feeling alone and rejected by others." Feeling alone and rejected seems to support, rather than contradict, the idea of isolation. Could you clarify how isolation contrasts with other experiences, such as being physically present with others but still feeling emotionally isolated? This distinction could highlight how individuals can be around people yet still experience a sense of disconnection. If this is not the intended meaning, it might be worth reconsidering whether the term "paradox" is the most appropriate choice in the original sentence. If isolation leads to feelings of loneliness and rejection, then it doesn’t necessarily present a paradox. Alternatively, is the paradox meant to convey that adolescents feel isolated when, in fact, they are not—perhaps due to their friends perceiving them as the ones pushing others away? This could suggest a distorted perception of isolation. In the discussion on line 516, you mention that "first, feelings of shame, burdensomeness and thwarted belongingness are characteristically experienced in the context of self-harm” and interestingly, they also contribute to social withdrawal and isolation. This implies a paradox where young people who self-harm experience thwarted belongingness yet also choose to withdraw socially. If this is the case, it would be beneficial to unpack this idea more clearly in the results section to clarify what you mean by "paradoxical."

DISCUSSION

• See major concerns point 3 and 4.

FIGURES

• Flow diagram – where it says "n=30" I initially interpreted that as you had posted 30 recruitment adverts. It could be clearer that it means 30 people responded to the advert? E.g., maybe put “Participants that responded to the recruitment advert posted in the research participant scheme, n =30” . Also not clear why some bits are capitilised e.g., Informed consent.

• Figure 3 – should “self-harm” be a box around the circles to show self-harm as the context? Just a thought. Do ignore though if this isn’t what you are trying to capture here.

SMALL COMMENT ON GRAMMAR

• I apologise if this comes across as pernickety, but there are some phrases in the manuscript that could be refined for clarity and consistency. For example, in line 7 of the abstract, you state, 'the present study conducted…' It might be clearer to say, 'in the present study, we conducted…' because attributing the conducting to the people rather than the study enhances clarity. If you do change this, be consistent throughout the manuscript.

I hope you find this feedback helpful, and I wish the authors continued success in their future work!

Reviewer #2: Thank you for the opportunity to review your manuscript. It was very clear and well-written and I believe the topic is relevant and important. You did a nice of describing the study, clearly articulating the findings, and connecting your findings to current literature. I have just a couple of minor suggestions that may help with clarity and consistency. In some places you use the term "adolescents" and in other places "young adults." I would stick to one term for clarity and consistency. Also, the language of the themes is slightly different in the abstract/results/discussion. I find it helps the same language is used throughout.

6. PLOS authors have the option to publish the peer review history of their article (what does this mean? ). If published, this will include your full peer review and any attached files.

**Do you want your identity to be public for this peer review?** For information about this choice, including consent withdrawal, please see our Privacy Policy .

Reviewer #1: No

Reviewer #2: No

---

## [Author Response · Author response to Decision Letter 1]

28 Jan 2025

Reviewer #1: OVERALL SUMMARY

Thank you for the opportunity to peer-review this important manuscript on the impact of friendships among young adults who self-harmed during adolescence. The paper addresses an underexplored topic in self-harm research, offering valuable insights into the interrelationship between friendships and self-harm, and the complexities involved.

The study’s findings emphasise that friendships not only influence self-harm behaviours but are also shaped by them, which is an important contribution to the field. The use of semi-structured interviews within a retrospective qualitative design, alongside a modified CaTS sort task and reflexive thematic analysis, constitutes a strong and appropriate methodology for capturing the subjective experiences of participants. It was particularly commendable to see the focus on both the onset and maintenance of self-harm in relation to friendships. The CaTS-AF task was used effectively to capture these temporal aspects, and the researchers made full use of this tool to enhance their analysis. Overall, the study identifies relevant and significant themes, and the interpretation of the findings is well-grounded, offering fresh perspectives on the connection between adolescent friendships and self-harm.

Thank you very much for your carefully considered, insightful and positive comments on our paper. We really appreciate your detailed constructive feedback on the research and its importance. Your suggestions have been extremely valuable in concisely addressing various issues, resulting in a clearer more robust report. Below we highlight the various changes made to address your concerns.

MAJOR CONCERNS

However, I have a few concerns that might be considered more “major”.

1) There needs to be greater consistency and clarity when discussing the types of friends involved in the study. It’s not always clear whether the focus is on same-age friends, online friends, or offline friends, as these distinctions are not consistently made in the methods or results. For example, the method section on line 136 notes that the interview schedule included questions about participants’ “offline and online friend relationships,” but in the results, there isn’t always clarification about which type of friend is being referred to, assuming participants specified this at all. At times, same-age peers are mentioned, but it’s not explicitly outlined whether this is the primary type of friend captured throughout the study. If the intention is to refer to "friends in general" unless otherwise specified, this should be made clear in the results, particularly since the majority of interviewees appear to discuss same-age friends. For example, in the discussion, the phrase "given their status as same-aged peers" (line 529) is used, but it’s unclear if participants consistently referred to same-aged peers throughout the study. This ambiguity could be addressed by more clearly specifying in the methods and results which types of friends—whether peers, online friends, offline friends, or same-age friends—are being discussed. If there were different types, it might be important to reflect on this in the discussion.

Thank you for the opportunity to clarify.

In the study, we allowed participants to self-identify for both friendships and self-harm, and we did not ask them to define or categorise their friendships (to avoid imposing restrictive definitions). As a result, the interviewer did not ask participants to define their friend context nor their friend characteristics (e.g., age, online vs offline) at any point throughout the interview. These were only known if participants explicitly mentioned such characteristics. Nonetheless, based on contextual information provided throughout each interview (i.e., indirect evidence), most interviews reference same-aged offline friends (most were from school and face to face, with some also being online and outside of school).

To address this, we have removed restrictive statements about types of friends discussed and we have clarified that most friendships are same-aged, school-based friendships.

REVISED. MANUSCRIPT (p.8, line 166): “offline/online friend relationships and the role of friends”

REVISED MANUSCRIPT (p. 12, lines 264-266): “Participants primarily discussed offline same-aged school-based friendships, with few mentions of online or older friends. Therefore, the following results mainly reference the former group unless otherwise specified.”

2) There is some inconsistency in the terminology used to describe the participants' developmental stages throughout the manuscript. Terms such as "adolescent" "young adult" "young person” are used. Given that adolescence is now often considered to extend up to age 24 (e.g., Sawyer et al. 2018 “the age of adolescence”), university students can still technically fall within this period. And given that the mean age of participants was 19 (with a range of 18 – 20) it might help to clarify that the developmental stage being discussed is still adolescence, but with self-harm recollected from early adolescence. This would ensure alignment with the current understanding of adolescence as extending into the mid-20s.

Thank you for the observation and we agree regarding inconsistencies. The different terminology as you rightly pointed out was mainly to differentiate between participants’ past experiences (in adolescence) and their present reflections (young adulthood). To ensure clarity regarding the terminology and the time period being discussed, we provide definitions of the specific age range considered in the introduction based on WHO definitions of adolescents and young people.

These terms are now used consistently throughout the updated manuscript as appropriate (e.g., if studies only include adolescent samples, we use the word adolescent; since our results mainly focus on adolescent experiences, we use the term adolescent in the results section). However, throughout the rest of the manuscript we have adopted more neutral terminology where possible (e.g., changing “adolescent” or “young people” for “participant” or “individual”):

REVISED MANUSCRIPT (p. 3, lines 36-38): “According to the World Health Organisation, adolescence encompasses the period from 10-19 years old, while the term young people refers to individuals between 10-24 years old (WHO, 2001).”

3) The limitations section could be expanded to address some important aspects that are currently missing. At present, the section is brief, only spanning two paragraphs, but there are several additional limitations that could be considered. For instance, the fact that the sample consists entirely of university students introduces an undergraduate population bias. The unique cultural and institutional context of the university may shape the participants' experiences and responses, and this should be acknowledged. While qualitative research is not focused on large sample sizes, but rather on the concept of data "saturation," it could still be valuable to address this explicitly in the limitations section. A brief comment on how saturation was approached or achieved would strengthen the methodological transparency of the study.

Thank you for your comment. We have now expanded the limitations section in order to include a more detailed reflection of limitations within the study. Based on your suggestions, we consider the relatively homogeneous sample in our study and how this may have impacted the conclusions drawn from the study.

As for data saturation, the approach we have utilised in the present study, namely Reflexive Thematic Analysis (Braun & Clarke, 2019), does not necessarily advocate for the use of data saturation. Instead, it assumes that themes and meanings are typically developed from the data collected and the process of interpretation (Braun & Clarke, 2021). Therefore, pre-establishing sample size is a difficult task when adopting this approach. For this, we considered typical recommendations for studies using RTA, participant availability/ participation, richness of the data collected, etc. This has been briefly reflected upon alongside potential limitations of our approach within the limitations section:

REVISED MANUSCRIPT (p., lines 686-701): “The present findings have certain limitations. First, despite the temporal view we adopted, we explored only a limited period, from early/mid-adolescence to young adulthood. Recent research highlights the potential for earlier self-harm onset (i.e. during childhood) and maintenance beyond young adulthood (Geoffroy et al., 2022). Nonetheless, our adolescent/young adult focus provides in-depth insight on a period of key relevance, given typical ages of self-harm onset, peak and decline (Whitlock & Selekman, 2014).

Second, the size and characteristics of our sample may have implications for the findings presented. The sample size was determined based on the overall research aim of providing an in-depth exploration of individuals experiences and sense-making. For this aim and when using an RTA approach, smaller samples of 10 to 20 participants are typically recommended (Braun & Clarke, 2019b). Despite the richness of the data captured in this study, it is important to reflect on the limitations of our purposeful sampling approach in sufficiently capturing participants’ experiences. Specifically, the sample consisted of undergraduate psychology students who may be a relatively homogeneous group with similar demographic and sociocultural characteristics and whose unique cultural and institutional context may have introduced potential population bias. As such and as mentioned in prior sections, the self-harm experiences of this sample may have certain similarities in trajectory and processes of sense-making. Therefore, it is plausible that certain self-harm experiences – where participants displayed balanced evaluations of their current situation in relation to both self-harm and friendships – were over-represented in our findings. Potentially, individuals with more difficult or unresolved experiences in the past and present chose not to participate, with implications for interpreting the current findings.”

4) Lastly, it would be useful to include further reflection on how the researcher’s position as a student may have influenced their interactions with participants, who were also students. This shared status could have affected the rapport between the researcher and participants, as well as the dynamics of the interviews. For instance, participants might have felt more comfortable or, conversely, may have withheld certain views based on perceived similarities or differences. On line 200, it is noted that a reflective diary was kept to account for the researcher’s impact during interviews and the analysis process, which is an important practice. However, it would be valuable to expand on this reflection in the discussion section. Even a brief mention of the specific insights or challenges arising from the researcher’s reflexive process could provide important context for interpreting the findings. If word count is a concern, this discussion could be summarised in one or two sentences within the main text, with more detailed reflections placed in the supplementary materials. A comment on reflexivity in the discussion, whether brief or expanded, would strengthen the study’s rigor by demonstrating an awareness of how the researcher’s positionality may have shaped the research process.

This is an excellent point. We have added a new section in the discussion based on your suggestion to capture our reflections around researcher’s positionality and its impact in the data collection/ interpretation process.

REVISED MANUSCRIPT (p. 28-29, lines 613-646):

“4.5 Reflexivity and researcher’s position

The use of qualitative methods inherently assumes that research is contextual and subjective (Dogson, 2019; Olmos-Vega, Stalmeijer, Varpio & Kahlke, 2023). This subjectivity and the researchers’ own positions, characteristics, experiences and their roles in shaping the research, should reflexively be considered (Macbeth, 2001; Dogson, 2019).

Particularly, my position as a doctoral student shaped the research in various ways across the different stages of the research process. First, being a student implied initial limited knowledge around the topic and how to best approach it. This informed the decision to allow participants to self-identify and describe their own self-harm and friendship experiences, without a priori constraints. This approach may have facilitated interviews by allowing participants to feel in control of the interview process and may have helped validate their experiences. Equally, this may have made the interview process more challenging in some cases due to the broad and non-restrictive nature of the questions. However, the use of the CaTS scaffolded the process for participants.

Additionally, similarities between myself and participants also shaped the research process and interpretation of findings. On one hand, being relatively similar in age and gender to most participants appeared to facilitate certain conversations and reduce power dynamics. In turn, it promoted relatability and shared understanding of the context in which certain self-harm experiences occurred (e.g., Tumblr, online friendships). On the other hand, similarities in age and educational background (i.e., psychology) may have shaped conversations in a particular way. During interviews, some participants appeared to use more clinical language and reasoning (e.g., conversations about warning signs/ symptoms of certain mental health disorders and how they manifested in themselves or others). Whilst this helped create mutual understanding, it also appeared to lead to more neutral and objective descriptions and sense-making, potentially detracting from more experiential aspects of participants’ accounts of their subjective experiences.”

MINOR SUGGESTIONS

Additionally, below are some minor suggestions specific to different sections of the manuscript, that may help to improve it further.

TITLE/ABSTRACT

• Title could better reflect the study by saying “role of friendships” e.g., [“People in my life just play different roles” A retrospective qualitative study of the role of friendships among young adults who self-harmed during adolescence”]

Thank you for your suggestion. We believe maintaining the original title ““People in my life just play different roles”: A retrospective qualitative study of friendships among young adults who self-harmed during adolescence” may be more suited in capturing the variety of themes in this study.

Whilst different friend “roles” or different friendships are discussed (as captured in the quote used for the title), the themes more broadly consider various aspects of friendships not limited to friends’ roles only (e.g., friendship characteristics, how SH influenced friendships).

• Abstract – lines 9/10 – might be better to have the (M=19.09; SD= 0.70; M=2, F=9) after the phrase young adults – because I read that as though the age they self-harmed was a mean age of 19.09 rather than that being the age of the sample.

Completely agree with this observation, thank you for noticing. This has now been corrected.

REVISED MANUSCRIPT (p.2, lines 9-10): “to explore the experiences of 11 young adults (M=19.09; SD=0.70; M=2, F=9) who self-harmed during adolescence.”

• Line 15 , could say “but they acquire” rather than “these acquire”.

This has now been corrected.

REVISED MANUSCRIPT (p.2, lines 15-16): “Furthermore, not only are friendships shaped by self-harm, but they acquire specific meanings…”

INTRODUCTION

• Line 56– maybe add in what friendships are defined as in the first place e.g., same age friends? Perhaps just outlining exactly what is meant by friendship in this context (this links in with the more ‘major’ concern comment 1 above).

Following your comment we have added a very broad definition of friendships and their roles during adolescence (in line 56). Additionally, we also explicitly clarify that friendships in this study mainly refer to in-person same-aged school-based friendships.

REVISED Manuscript (p. 4, line. 56-64): “Friendships are typically consi

---

## [Editor Report · Decision Letter 1]

16 Feb 2025

“People in my life just play different roles” A retrospective qualitative study of friendships among young adults who self-harmed during adolescence

PONE-D-24-36301R1

Dear Dr. Bilello,

Thank you for submitting your revised manuscript and the careful responses to the reviewers' comments on your paper. We are pleased to inform you that your manuscript has been judged scientifically suitable for publication and will be formally accepted for publication once it meets all outstanding technical requirements.

Within one week, you will receive an e-mail detailing the required amendments. When these have been addressed, you will receive a formal acceptance letter and your manuscript will be scheduled for publication.

Kind regards,

Professor Sriya Iyer

Academic Editor

PLOS ONE
---

## [Editor Report · Acceptance letter]

PONE-D-24-36301R1

PLOS ONE

Dear Dr. Bilello,

I'm pleased to inform you that your manuscript has been deemed suitable for publication in PLOS ONE. Congratulations! Your manuscript is now being handed over to our production team.

Kind regards,

on behalf of

Professor Sriya Iyer

Academic Editor

PLOS ONE